# Optical wireless link between a nanoscale antenna and a transducing rectenna

Arindam Dasgupta[1], Marie-Maxime Mennemanteuil[1], Mickaël Buret[1], Nicolas Cazier[1], Gérard Colas-des-Francs[1] & Alexandre Bouhelier[1]

Initiated as a cable-replacement solution, short-range wireless power transfer has rapidly become ubiquitous in the development of modern high-data throughput networking in centimeter to meter accessibility range. Wireless technology is now penetrating a higher level of system integration for chip-to-chip and on-chip radiofrequency interconnects. However, standard CMOS integrated millimeter-wave antennas have typical size commensurable with the operating wavelength, and are thus an unrealistic solution for downsizing transmitters and receivers to the micrometer and nanometer scale. Herein, we demonstrate a light-in and electrical signal-out, on-chip wireless near-infrared link between a 220 nm optical antenna and a sub-nanometer rectifying antenna converting the transmitted optical energy into direct electrical current. The co-integration of subwavelength optical functional devices with electronic transduction offers a disruptive solution to interface photons and electrons at the nanoscale for on-chip wireless optical interconnects.

[1] Laboratoire Interdisciplinaire Carnot de Bourgogne, CNRS UMR 6303, Université de Bourgogne Franche-Comté, 9 Avenue A. Savary, 21000 Dijon, France. Correspondence and requests for materials should be addressed to A.B. (email: alexandre.bouhelier@u-bourgogne.fr)

Conventional radiowave and microwave antennas operate a bilateral energy conversion between electrical signals and electromagnetic radiation. Integration of these antennas to modern consumer electronics is thus widely adopted for long-range and short-range data transfer. Vivid examples are communication between mobile devices and remote biosensors for healthcare providers[1,2]. Similar sub-wavelength interfacing of optics and electronics is envisioned as an archetype to reduce the size of communication devices while maintaining their necessary speed and bandwidth[3,4]. This is a daunting task as state-of-the-art electronics is unable to respond to the fast alternating fields associated with optical frequencies and the size of photonic components remains orders of magnitude larger than their electronic counterparts. Nanoscale antennas operating in the optical domain are technologically available, but have been mainly limited to interfacing near-field and far-field radiation by tailoring the momentum of light[5].

In 2010, A. Alù and N. Engheta theoretically proposed an optical wireless channel between two nanoscale antennas[6]. The idea was partially demonstrated by beaming either an optical signal towards a distant luminescent receiver[7] or by the mediation of surface plasmons[8]. Despite an optimization with highly directive antennas[9,10], the link operates on the basis of a light-in and light-out configuration without transduction of the transferred optical energy to an electronic signal. Recent progress shows that tunneling nonlinearity in the conduction of an atomic scale tunnel junction can rectify the plasmonic response at optical frequencies to direct current (d.c.)[11–15]. Immediately, these rectifying antennas, or rectennas, appear to be essential ultrafast devices for merging optics and electronics at the nanoscale.

Here, we exploit these functionalities on a single platform to realize an optical wireless power transmission between laser-illuminated nanoscale dipolar antennas and a distant rectenna. We demonstrate a proof-of-principle wireless link where gold nano-antennas act as transmitters by directing an out-of-plane laser signal towards an optical rectenna where the transmitted radiation is rectified as d.c. current. We show that the rectified current is governed by the polarization of laser illuminating the distant transmitter antenna. The amount of transmitted power through the wireless link is correctly described by the Friis transmission equation and follows an inverse square relation with the distance between the transmitting antenna and the rectenna. The findings experimentally confirm that an optical signal can be transmitted and transduced between two nanoscale units without relying on a physical optical waveguide.

## Results

**Device principle.** Figure 1a illustrates the optical line-of-sight channel presented here. The transmitter is a laser-illuminated gold nanodisc acting as an optical dipole antenna. The radiation scattered by the antenna is remotely detected and converted to a d.c. current by a distant electrically biased rectenna. A transmission optical image of the units is shown in Fig. 1b. The Au electrodes powering the rectenna feed-gap and a series of optical antennas (black dashed box) are readily seen with a dark contrast. The tunneling feed-gap of the rectenna is formed by electromigration of initially touching electrodes[16]. Figure 1c is a false colored scanning electron microscopy (SEM) image of the white dotted box in Fig. 1b showing the in-plane tunneling junction together with an adjacent optical antenna. The electrical characterization confirms that tunneling is driving electron transport (see Methods section). The optical antennas have a fixed diameter of 220 ± 10 nm to simultaneously induce a dipolar response and maximize the scattering efficiency (see Supplementary Figure 8).

The devices are immersed in a refractive index (RI) matching oil to operate the link in a homogenous environment.

**Optical rectification.** Let us assume an optical rectenna formed by a tunnel junction where the conduction mechanism remains the same over a pulsation range $2\omega_{ac}$. A d.c. current $I(V_{dc})$ is tunneling through the feed-gap when a bias of $V = V_{dc}$ is applied between the two metal leads. If a small a.c. bias $V_{ac}$ of frequency $\omega_{ac}$ is superposed to $V_{dc}$, the total current through the junction can be expressed by a Taylor's expansion[17],

$$
\begin{aligned}
&= I(V_{dc}) + \frac{\partial I}{\partial V}\Big|_{V_{dc}} V_{ac}\cos\omega_{ac}t + \frac{1}{2}\frac{\partial^2 I}{\partial V^2}\Big|_{V_{dc}} V_{ac}^2\cos^2\omega_{ac}t + \cdots \\
&\approx \left[ I(V_{dc}) + \frac{1}{4}\frac{\partial^2 I}{\partial V^2}\Big|_{V_{dc}} V_{ac}^2 \right] \\
&\quad + \frac{\partial I}{\partial V}\Big|_{V_{dc}} V_{ac}\cos\omega_{ac}t + \frac{1}{4}\frac{\partial^2 I}{\partial V^2}\Big|_{V_{dc}} V_{ac}^2\cos 2\omega_{ac}t + \cdots
\end{aligned}
\tag{1}
$$

The time-independent term in Eq. 1 indicates the presence of an additional rectified current $I''$, which is proportional to the nonlinearity of the conductance $\partial^2 I/\partial V^2$ and $V_{ac}^2$: $I'' = 1/4 V_{ac}^2 (\partial^2 I/\partial V^2)$.

To describe optical rectification, a quantum mechanical treatment is generally used[18]. When the rectenna is illuminated with light of energy, the response builds up an a.c. voltage $V_{opt}$ of pulsation $\omega$ across the junction. This optical potential triggers a photon-assisted tunneling of electrons to produce a d.c. photocurrent in addition to the $I(V_{dc})$. If $eV_{opt} \ll \hbar\omega$, the rectified d.c. photocurrent is given by[19,20]

$$
\begin{aligned}
I_{phot} &= I\left(V_{dc}, V_{opt}, \omega\right) - I(V_{dc}) \\
&= \frac{1}{4}V_{opt}^2\left[\frac{I(V_{dc}+\hbar\omega/e) - 2I(V_{dc}) + I(V_{dc}-\hbar\omega/e)}{(\hbar\omega/e)^2}\right]
\end{aligned}
\tag{2}
$$

For gold and for an excitation energy $\hbar\omega < 2$ eV, the tunneling transmission remains smooth within the range $E_F \pm \hbar\omega$, where $E_F$ is the Fermi energy[11]. Therefore, Eq. 2 reduces to its classical form $I_{phot} = 1/4 V_{opt}^2(\partial^2 I/\partial V^2)$ with $V_{opt}$ playing the role of $V_{ac}$ and $I_{phot}$ of $I''$[11,21].

We use a low frequency a.c. voltage and lock-in detection to record the conductance $G = \partial I/\partial V$, the current proportional to the nonlinearity $\partial^2 I/\partial V^2$ and the laser-induced current $I_{phot}$[11,12,17]. The description of the complete measurement system is included in the Methods section (see Supplementary Fig. 1 for the schematic of the experimental set-up). The plot of conductance of the junction $G(V_{dc})$ in Fig. 1d features a zero bias conductance of about 10 µS ($\approx 0.13\ G_0$ where $G_0 = 77.5$ µS is the quantum of conductance). Using Simmons' model of tunneling transport[22,23], we qualitatively estimate the feed-gap width to be <0.5 nm (Supplementary Fig. 2, Supplementary Fig. 3 and Supplementary Note 1). In Fig. 1e, we plot the bias dependence of $I''$ (line) and $I_{phot}$ (points) for three laser intensities when the illumination is directly focused on the rectenna feed-gap. For each excitation intensity, $V_{ac}$ is adjusted to obtain $I_{phot} = I''$. Under this condition, the optical voltage generated across the feed-gap equals the low frequency voltage applied between the electrodes[11]. From the shared voltage dependence of $I_{phot}$ and $I''$, we conclude that the photocurrent generated at the rectenna results from optical rectification. Additional experiments to rule out any thermal contributions are presented in Supplementary Note 2 and Supplementary Fig. 4.

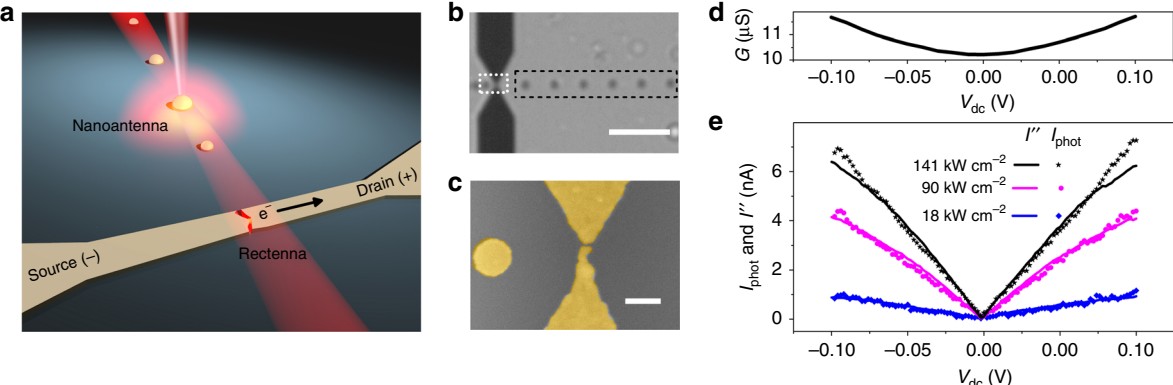

**Fig. 1** Concept and device characterization. **a** Schematic illustration of the optical wireless transducing link: a dipolar optical antenna is excited by a focused laser beam. The energy radiated by the transmitter antenna is detected and converted to an electrical signal by a distant electrically biased receiving rectenna. **b** Optical transmission image of the functional units. The white dotted box highlights the feed-gap of the rectenna and the black dotted frame contains a series of transmitter optical antennas placed at different distances from the rectifying gap. Scale bar is 4 μm. **c** Colorized SEM image of the region highlighted by the white dotted rectangle in **b** featuring a rectifying nanoscale feed-gap between two gold electrodes. Gold is colorized in yellow. Scale bar is 250 nm. **d** Evolution of the conductance $G$ of the tunnel feed-gap with $V_{dc}$. **e** Plot of the variation of the amplitude of the rectified photocurrent $I_{phot}$ (data points) and of the current proportional to the nonlinearity of the junction's conductance $I'' = 1/4(V_{ac}^2 \partial^2 I / \partial V^2)$ (line plots) as a function of applied bias $V_{dc}$ and for three direct excitation intensities of the rectenna illuminated by a focused 785 nm laser. The shared trends between $I_{phot}$ and $I''$ confirm an optical rectification mechanism

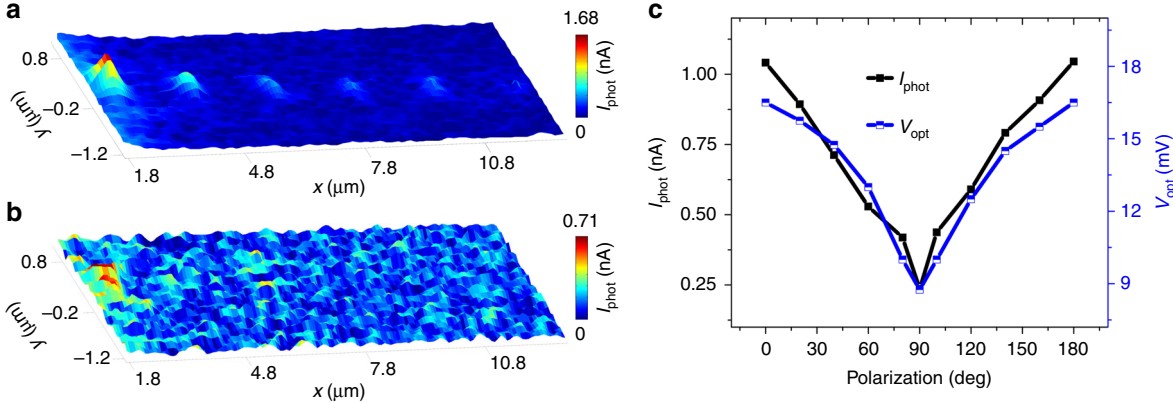

**Fig. 2** Polarization response of the optical link. **a** Photocurrent map generated by the rectenna reconstructed pixel-by-pixel by scanning the optical antennas through the laser focus. The scanned region is indicated by the black dotted rectangle in Fig. 1b. The incident polarization is along the vertical direction, $y$-axis (0°) for excitation intensity 353 kW cm$^{-2}$. **b** The same region for an incident polarization along the horizontal direction, $x$-axis (90°). $V_{dc}$ is constant at 50 mV. **c** Plot of $I_{phot}$ and $V_{opt}$ as a function of incident polarization for individual excitation of an antenna which is 4 μm away for an excitation laser intensity of 540 kW cm$^{-2}$

**Optical wireless link**. In the following section, we assess the operation of a wireless link when a remote optical antenna is broadcasting a signal towards this transducing rectenna. The laser is no longer directly incident on the rectenna, but is focused alternatively on the series of optical antennas displayed in Fig. 1b. Figure 2a shows a pixel-by-pixel reconstructed photocurrent map generated by the rectenna. The laser excitation is polarized along $y$-axis (0°) and the sample is scanned through the focal area (step size 70 nm). $V_{dc}$ across the rectenna is fixed at 50 mV and the laser intensity at 353 kW cm$^{-2}$. The important conclusion drawn from the current map is the presence of a rectenna response whenever the laser excites a remote optical antenna. The response is observed even for separation distances exceeding several micrometers. When the polarization is turned by 90° ($x$-axis), the photocurrent generated at the rectenna vanishes nearly completely (Fig. 2b). To confirm the transduction of the signal radiated by the optical antennas, we station the laser on the optical antenna located 4 μm away from the rectenna (2nd antenna from

the right in Fig. 1b) and simultaneously monitor $I_{phot}$ and $I''$ as a function of $V_{dc}$ while rotating the polarization. The intensity of the laser is 540 kW cm$^{-2}$ for this experiment and we verified that for all the polarizations angles, $I_{phot}$ follows $I''$ proving thus the $I_{phot}$ signal is optically rectified (Supplementary Fig. 6 and Supplementary Note 3). The value of $V_{ac}$ for conditioning $I_{phot} = I''$ is recorded as a measure of the optically induced a.c. voltage $V_{opt}$ at the feed-gap. The evolution of $I_{phot}$ and $V_{opt}$ as a function of the incident laser polarization is plotted in Fig. 2c for a bias $V_{dc} = 50$ mV. It is clear that the rectification process is suppressed as the polarization is rotated by 90°. This polarization dependence of the current produced by the rectenna removes any concern about a heat-mediated transduction mechanism since the temperature of the illuminated disc antenna is driven by the absorption coefficient and not by the polarization of the incident laser beam. As an additional control experiment we map d$I$/d$V$ at the frequency of the optical chopper (Supplementary Fig. 5). No measurable contrast can be related to the photocurrent map presented in

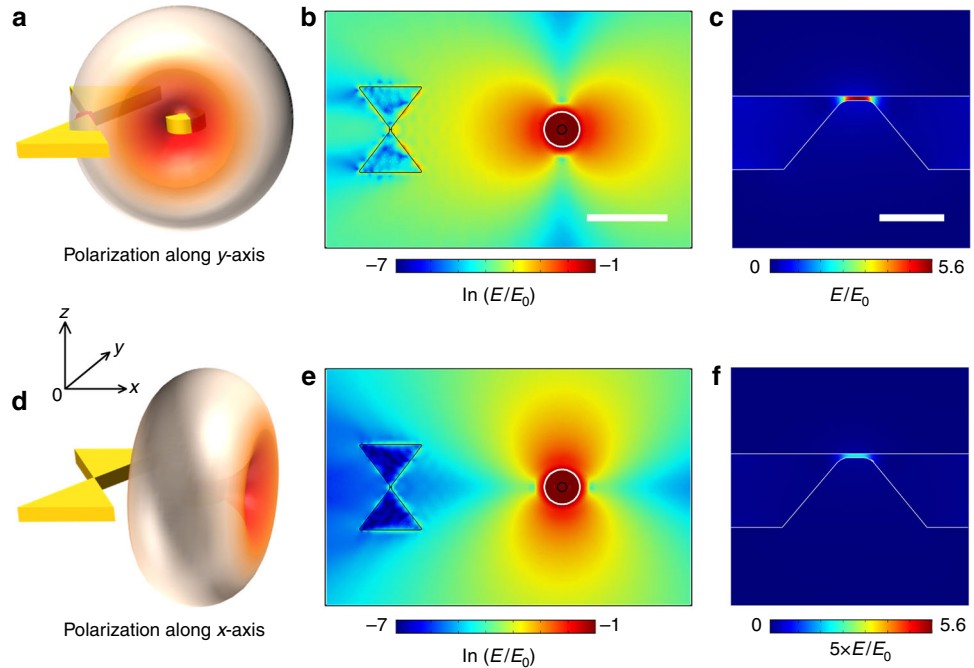

**Fig. 3** Numerical simulations of the polarization dependence of the wireless power transfer. **a** Schematic of the operation principle of the wireless link. The rectenna is modeled as two Au triangles separated by a 0.5 nm gap. The optical antenna is a 220 nm Au antenna located 4 μm to the right of the rectenna (black circle). When the transmitting antenna is illuminated with a laser polarized along y-axis, the dipolar radiation is directed towards the receiving antenna. **b** Calculated distribution of the electric field, plotted in logarithmic scale, for an incident polarization along y-direction. The white circle indicates the size of the excitation in the form of a focused Gaussian beam where the focal spot is considered to be 600 nm in diameter. Scale bar is 2 μm. **c** Zoomed image of the electric field distribution present at the feed-gap of the rectenna. Scale bar is 10 nm. **d** Same configuration with an incident polarization along x-axis. The disc radiates in the orthogonal direction and hence the rectenna receives minimum amount of the transmitted energy. **e** and **f** are the electric field maps for an incident polarization along the x-axis. For clarity, the electric field in **f** is multiplied by a factor 5. The electric field created at the gap decreases drastically when the signal emitted by the antenna is not directed toward the rectifying feed-gap

Fig. 2a (more discussion in Supplementary Note 2) indicating that the gap size is not deformed by the heat produced at the distant antenna[24]. We reproduce the wireless link and the transduction of the optical signal with a second tested device (Supplementary Fig. 7) and assess the main parameter affecting the device-to-device variability in Supplementary Table 1 and Supplementary Note 4.

**Numerical modeling**. Numerical simulations based on a three-dimensional finite element method (3D-FEM) bring an understanding of the polarization dependence of the rectified signal. The rectenna is modeled by two truncated Au triangles separated by a distance of 10 nm except at the middle where a small protrusion on the bottom electrode reduces the gap size to 0.5 nm. A 220 nm diameter Au disc antenna is placed 4 μm away from the feed-gap. The entire geometry is placed inside a homogeneous medium of RI = 1.52. When excited by a linear polarization at 785 nm, the disc behaves as a dipolar resonant antenna radiating its characteristic two-lobe pattern perpendicularly to the incident electric field. Figure 3a, d are schematic representations picturing the dipolar radiation for two orthogonal in-plane polarizations. Figure 3b is the calculated electric field distribution plotted in logarithmic scale when the antenna is illuminated (the white circle indicates the excitation area) with a polarization along the y-axis (vertical). Clearly, the optical antenna redirects the far-field radiation towards the rectenna feed-gap. The interaction of this receiving radiation with the tunneling gap results in a high electric field at the junction[25], enhanced by a factor of approximately 5.6 compared to the excitation field, which is illustrated in the zoomed-in electric field map in Fig. 3c. When the dipolar radiation is polarized perpendicularly to the receiver (Fig. 3d), the

field enhancement at the junction is reduced to 0.44 (Fig. 3e, f), explaining the vanishing photo-response of Fig. 2d. More details about the modeling procedure are included in the Methods section.

**Controlling the efficiency of the wireless link**. When a transmitter and a receiver constituting a wireless link are separated by a distance d, the received power is given by the Friis equation[26],

$$P_{\mathrm{r}} = \left[\eta_{\mathrm{r}}\eta_{\mathrm{t}}D_{\mathrm{r}}D_{\mathrm{t}}\left(1-|\Gamma_{\mathrm{r}}|^2\right)\left(1-|\Gamma_{\mathrm{t}}|^2\right)|a_{\mathrm{r}} * a_{\mathrm{t}}|^2 \frac{\lambda_{\mathrm{ex}}}{4\pi d^2}\right]P_{\mathrm{fed}} \quad (3)$$

$P_{\mathrm{fed}}$ is the power fed to the transmitter, $\lambda_{\mathrm{ex}}$ is the operational wavelength, $\eta_{\mathrm{r}}$ and $\eta_{\mathrm{t}}$ are the radiation efficiencies, $D_{\mathrm{r}}$ and $D_{\mathrm{t}}$ are the directivities, $\Gamma_{\mathrm{r}}$ and $\Gamma_{\mathrm{t}}$ are the reflection coefficients, $a_{\mathrm{r}}$ and $a_{\mathrm{t}}$ represent the polarizabilities of the receiving and the transmitting antennas, respectively.

We first analyze the distance dependence of the optical wireless link. The evolution of the amplitude of the rectified current $I_{\mathrm{phot}}$ is plotted in Fig. 4a as a function of the distance d separating the antenna from receiving rectenna. $I_{\mathrm{phot}}$ (black data points) is fitted with a generic power law function $\alpha d^b + c$ where $\alpha$ represents the coupling strength between the transmitter and the receiver and c is the dark photocurrent of the device ($\approx 0.32$ nA). The best fit gives an exponent $b = -1.8$, which is close to the expected inverse square law dependence (Eq. 3). The slight mismatch may be due to the inevitable deviations in the antenna geometry, and misalignment. We also record the evolution of $V_{\mathrm{opt}}$ as a function of d, which is shown as the magenta plot in Fig. 4a. This is done by focusing the laser individually on each antenna and varying

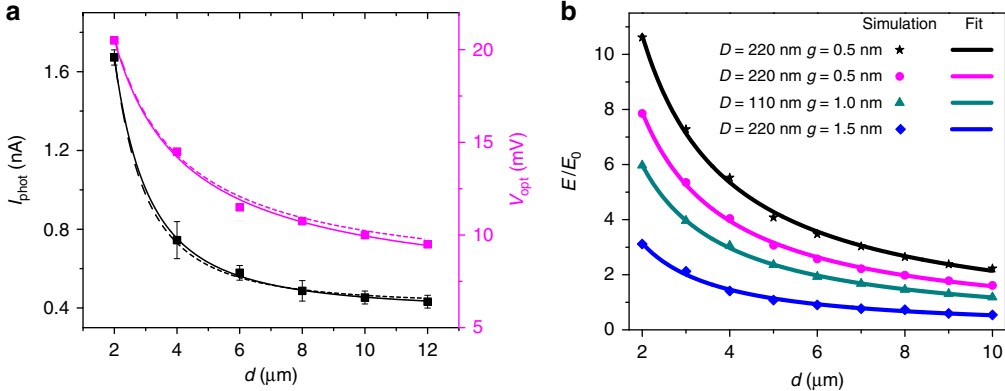

**Fig. 4** Characteristics of the optical wireless power transfer. **a** Dependence of $I_{phot}$ and $V_{opt}$ on the distance between the illuminated optical antenna and the rectenna for an excitation intensity of 352 kW cm$^{-2}$ (black and magenta squares, respectively). The solid black and magenta lines indicate the best fits obtained by using a generic power law function ($\alpha d^b + c$). For comparison, $d^{-2}$ and $d^{-1}$ dependences for $I_{phot}$ and $V_{opt}$ are also shown as black and magenta dashed lines, respectively. **b** Simulated results of the electric field amplitude at the feed-gap vs $d$ for different combinations of gap widths and antenna diameters. The power law fits (solid lines) always converge to an exponent close to −1. The field amplitude decreases with the gap width as well as for an off-resonant antenna ($D = 110$ nm)

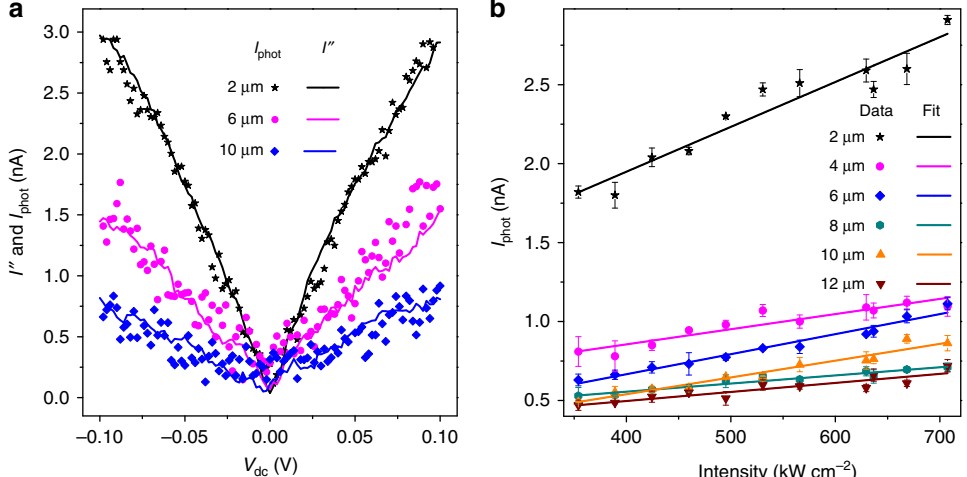

**Fig. 5** Transduced current vs bias and excitation laser power. **a** Variation of $I_{phot}$ and $I''$ as a function of $V_{dc}$ for three distant optical antennas. **b** Plot of $I_{phot}$ as a function of excitation laser intensity for all the antennas. The data are recorded for $V_{dc} = 50$ mV

$V_{ac}$ to equalize $I_{phot}$ and $I''$ for a sweep of $V_{dc}$ across −0.1 V to 0.1 V (see Fig. 5a). The evolution of $V_{opt}$ with the distance $d$ is also fitted with a power law with a similar expression. Here $c$ again indicates the noise level, which is here $c = 6.5$ mV. The best fit gives $b = −0.85$ (magenta solid plot), which is close to −1. This is expected, as $I_{phot}$ is proportional $V_{opt}^2$ (Eq. 2).

We corroborate our experimental distance dependence of the rectified current and optically induced voltage drop at the junction (Fig. 4a) with 3D-FEM based calculation of the electric field created at the rectenna's feed-gap for varying distance $d$ between the rectenna and the transmitting unit. The normalized electric field produced at a gap size of 0.5 nm and a 220 nm transmitter antenna is presented in Fig. 4b (black data points). The power law fit to the distance dependence (black line) converges to an exponent close to −1, which agrees with the experimental dependence of $V_{opt}$.

As indicated in Eq. 3, the amount of power transferred not only depends on the distance, but also on parameters that are influenced by the geometry of the antenna[5,27]. We thus numerically explore the influence of the gap size $g$, and the effect of an off-resonant antenna. Figure 4b indicates that sub-nm separation favors a large electric field at the feed-gap. However,

our calculations do not take into account quantum-size effects[28], which may limit the amplitude of the electrical field. Broadcasting the optical signal with a non-resonant antenna of diameter $D = 110$ nm also leads to a reduced electrical field at the transducing feed-gap (Supplementary Fig. 8 and Supplementary Note 5). We verify the minimal influence of the presence of other antennas in the transmission path in Supplementary Fig. 9 and Supplementary Note 5.

**Transduction yield**. We estimate the overall transduction yield $\kappa$ of the link discussed here. $\kappa$ is a measure of what fraction of the incident laser power is converted to electrical power through rectification and is given by the following equation:

$$\kappa (\text{in dBm}) = 10 \times \log \left[ \frac{I_{phot}^2}{G \times (1 \text{ mW of incident power})} \right] \quad (4)$$

where $G$ is the conductance of the tunnel junction.

We estimate $\kappa = −91$ and $−101$ dBm for the radiation transmitted by the antennas situated 2 and 10 μm away respectively for $V_{dc} = 50$ mV applied across the junction.

**Table 1 Coupling strength between the optical nanoantennas and the rectenna**

| Optical antenna | Coupling strength (nA cm$^2$ kW$^{-1}$) |
|---|---|
| Transmitting from 2 μm | $2.84 \times 10^{-3}$ |
| Transmitting from 4 μm | $9.59 \times 10^{-4}$ |
| Transmitting from 6 μm | $1.27 \times 10^{-3}$ |
| Transmitting from 8 μm | $5.15 \times 10^{-4}$ |
| Transmitting from 10 μm | $1.06 \times 10^{-3}$ |
| Transmitting from 12 μm | $5.73 \times 10^{-4}$ |

The transduction yield improves with increasing the applied d. c. bias to the junction. This is observed in the experimentally recorded $I_{phot}$ vs $V_{dc}$ plots for three distant antennas presented in Fig. 5a. An increase of $\kappa$ is indeed expected as the nonlinearity of the conductance of the rectenna increases as we raise $V_{dc}$ (see the evolution of $G$ in Fig. 1d). For the 2 μm distant antenna, setting $V_{dc}$ to 100 mV improves the transduction yield to −87 dBm. Note that here $\kappa$ denotes the overall efficiency of the link including antenna radiation, signal propagation, and transduction. In radio-frequency wireless communication, the received signal strength indicator (RSSI) evaluates the quality of the transmitted signal measured at the reception node before transduction. A transmission channel with a RSSI comprised between −70 and −100 dBm is considered as fair for basic connectivity.

Finally, we plot in Fig. 5b the dependence of measured rectified photocurrent on the excitation laser intensity for each remote transmitting antenna for $V_{dc} = 50$ mV. Regardless of the distance, the photocurrent scales linearly with the signal intensity fed to the optical antennas in agreement with Eq. 3. The slopes of the linear fits give the coupling strengths between each transmitter and the rectenna, which is measure of the change in the transduced currents per kW cm$^{-2}$ increment in the illumination intensity. The values are reported in Table 1. In Supplementary Table 2, we compare the photocurrent produced by a direct illumination of the rectenna with the photocurrent generated via the mediation of distant optical antennas for an illumination intensity 353 kW cm$^{-2}$ and $V_{dc} = 50$ mV.

## Discussion

In conclusion, we demonstrate an on-chip nanoscale optical wireless link between a laser-illuminated optical antenna and a transducing rectenna. In our experiments, simple gold nanodiscs act as a polarization-sensitive transmitting antenna directing the laser radiation towards the rectifying gap antenna. The amount of power transferred maintains an inverse square relation with the distance between the transmitting antenna and the rectenna. In addition to this, the geometrical properties of the participating units contribute to the yield of power transfer. Further improvement in the efficiency of transmission can be achieved by integration of highly directional optical antennas and resonant feeds.

An integrated wireless transmission enables a new communication strategy between nanoscale devices. This can be deployed when physical links (e.g., integrated photonic waveguides) cannot be implemented or when the transmitted signal should be outcoupled via the mediation of optical antennas and metasurfaces[29,30]. The wireless link can immediately be applied to develop ultrafast electro-optical connectors. Transmission rates in excess of $10^{12}$ bits per second are achievable as the plasmonic response of the gap is defined by the polarizability response of the metal and not by carrier lifetime as in semiconductors[31]. Recent progress shows that an electrically driven tunneling junction can act as ultrafast broadband self-emitting device[32–34]. Therefore, integration of a wireless link between an electron-fed optical antenna with a transducing rectenna may enable ultrafast

information transfer. Such devices will represent a paradigm for on-chip interfacing of electrons and photons at the nanoscale.

## Methods

**Sample preparation**. The samples are prepared on a glass coverslip by a double step lithography involving electron beam lithography (EBL) and photolithography. The antennas (disc of diameter 220 ± 10 nm), a nanoscale constriction of length 400 nm and width 100 nm bridging two large triangular Au structures as well as alignment marks are fabricated during a first step by EBL. For this, we spin coat and bake a 400 nm thick double-layer of poly(methyl-methacrylate) (PMMA) constituted of two different molecular weights (50 and 200 kDa). We then sputter a sacrificial gold conductive layer (5–10 nm) on the resist layers to avoid charging during electron beam exposure. Once the PMMA is exposed and developed, a 2 nm thick Ti adhesion layer followed by a 45 nm thick Au are subsequently deposited through thermal evaporation. The excess metal is then lifted off to obtain the final nanostructures. Then the macroscopic gold electrodes contacting the triangular pads are realized via standard optical lithography. During this step, the alignment marks are used to position the sample coordinates with respect to the coordinate system of the photolithography mask. Once the nanostructrures and electrical connections are fabricated, we produce the nanoscale gap-antenna by controlling the electromigration of the constriction. The electromigration is stopped once the zero-bias d.c. conductance reaches a value lower than the quantum conductance ($G_0 = 77$ μS). The presence of a tunnelling gap is confirmed by measuring the nonlinear I–V characteristics of the device (see Supplementary Fig. 2).

**Electrical and optical measurements**. The schematic of the experimental setup we use for the optical and electrical characterizations presented in the paper is illustrated in the Supplementary Fig. 1. With the help of an inverted optical microscope (Nikon, Eclipse), we focus the 785 nm wavelength laser on the sample through an oil immersion objective lens of numerical aperture (NA) 1.49. During the whole experiment, nanostructures are immersed in RI matching oil of RI = 1.52. For electrical measurements, the sample is biased with an applied d.c. bias $V_{dc}$ added with a small modulation a.c. voltage $V_{ac}\cos\omega_1 t$ at $V_{ac} = 20$ mV where $f_1 = \omega_1/2\pi = 12.37$ kHz is the modulation frequency. A first lock-in amplifier (Zurich Instrument, HF2LI) referenced at $f_1$ and $2f_1$ is used to simultaneously record the first harmonic ($\partial I/\partial V$) and second harmonic (1/4 $V_{ac}^2 \partial^2 I/\partial V^2$), of the current tunneling through the gap. The first harmonic is proportional to the conductance and second harmonic signifies nonlinearity in the junction's conductance. To extract the laser-induced current $I_{phot}$ the laser beam is chopped by a chopper (Thorlabs) at frequency $f_{chop} = 831$ Hz and the tunnelling current is demodulated at $f_{chop}$ using a second lock-in amplifier (Zurich Instrument, HF2LI). Therefore with this arrangement, $I(V_{dc})$, $\partial I/\partial V$, $\partial^2 I/\partial V^2$ and $I_{phot}$ can be measured simultaneously as a function of $V_{dc}$. For mapping, the sample is scanned through the laser spot by the moving the sample with a linearized piezoelectric stage (PI, P-545) and the data are recorded by a scanning data acquisition system (RHK, R9).

**Numerical modeling**. We have performed the 3 dimensional-finite element method (3D-FEM) modeling of the system using a commercial FEM solver COMSOL Multiphysics. We use the RF module of the solver to perform all the simulations. For simplicity, the rectenna is modeled as a gap junction between two 40 nm thick Au triangles. The gap width is 10 nm except at the center where a small protrusion on the bottom electrode reduces the gap size to 0.5 nm. The optical antennas are modeled by 220 nm diameter 40 nm thick Au cylinders. The electrical permittivity of Au is taken from the experimental values provided by Johnson and Christy[35]. The entire geometry is placed inside a homogeneous medium of RI = 1.52. The excitation is a 785 nm focused Gaussian beam incident on the antenna. The focal spot of the excitation beam is 600 nm diameter. This corresponds to an intensity full-width at half-maximum comparable to the point-spread function of the objective (i.e., 320 nm). We then calculate the electric field distribution in the computation window encompassing the transmitting antennas and the rectenna. We use perfectly-matched layer at the boundary to mitigate spurious reflections.

**Data availability**. Data are available from the corresponding author upon reasonable request.

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

## Acknowledgements

This work has been supported by the European Research Council under the European Community's Seventh Framework Program FP7/ 2007–2013 Grant Agreement No. 306772. Device fabrication was performed in the technological platform ARCEN Carnot with the support of the Région de Bourgogne. We thank I. Smetanin, O. Demichel for discussions, J. Dellinger for its initial implication in the setup, S. Pernot, and B. Sinardet for developing part of the electronic unit and L. Novotny for an initial reading of the manuscript.

## Author contributions

A.D. conducted the experiment, analyzed the data, and performed the simulations. M.M. M. developed the experimental and fitting procedures, N.C. and M.B. constructed the measurement apparatus to control the electromigration process, G.C.D.F. supervised the simulations. A.B. conceived the experiment and supervised the research. A. D. and A. B. wrote the manuscript, with revisions by all.

## Additional information

**Competing interests:** The authors declare no competing interests.

