## [Peer Review File · Nature Communications]

Reviewers' comments:

Reviewer #1 (Remarks to the Author):

The authors report the use of the optical rectification in an electromigrated nanogap to detect radiation scattered by nearby plasmonic disks. They establish that optical rectification is the most plausible mechanism responsible for the photocurrent through its dependence on dc bias and comparison with (effectively) dc transport, as well as its dependence on the laser position. The authors find that the spatial and polarization dependence of the (dc) photocurrent are consistent with computational modeling of the optical scattering of the plasmonic disks.

I find the experimental data and numerical modeling to be quite persuasive. I do have a few comments that I think the authors need to address prior to publication:

- 1) How many device structures have shown this clear response? That is, unless I'm missing something, it seems that all of the data in the paper and the supporting material come from one particular electromigrated junction. Is that correct? Reproducibility is important, as is establishing the amount of device-to-device variability. One would expect the efficiency of the transduction between excited disk and nanogap to depend sensitively on the nanogap electrical nonlinearity; it would be useful to see this comparison.
- 2) The authors have already performed another cross-check that measured photocurrent response is not dominated by some thermal response. The absorption and hence heating of the illuminated disks should be insensitive to the incident polarization. The fact that the photocurrent does have polarization dependence (consistent with the modeling), while heating due to the disks should be largely polarization independent, is additional evidence in favor of the authors' interpretation of the mechanism.
- 3) The authors should say something in the methods or the supporting material about how they performed the finite element simulations. What software was used? What dielectric functions were assumed for the metal? Did they include the Ti or TiOx in the calculations? Was the incident excitation modeled as a gaussian beam, or was the incident excitation modeled as a plane-wave? I would think the former, but it is important to make this clear to the reader.
- 4) It would be helpful for the authors to calculate an efficiency of the coupling process. The optical rectification in these nanogap structures has historically had a very poor efficiency in terms of power. If there really is to be an application of similar structures in on-chip signal processing, efficiency is important to quantify.
- 5) I would suggest that the authors qualify some of the statements made in the last few sentences of the paper. For example, better to say something like "in principle, data rates as high as 10^{12} bps could be possible, as the ultimate limiting factor is the polarization response of the metal". (Note also that the authors must mean 10^{12} bps, not 10^{12} Mbps!) Also, the statement that the enhanced field in the nanogap is appropriate for driving transistor operations seems not to be well motivated. How would that work? Better to leave that out unless there is some clear reasoning.

In summary, I find the data and interpretation to be very interesting and persuasive. I think that the paper could be quite influential in the field, but it is important to establish reproducibility and to provide further quantification of efficiency, as well as greater information about the numerical modeling.

Reviewer #2 (Remarks to the Author):

Summary:

The authors report on a configuration of energy transmission via an in-plane wireless link and the detection by electrical rectification in an atomic scale tunneling junction. A systematic analysis of the device is presented on the current generated when the junction is illuminated directly by out-of-plane and in-plane sources. Interestingly, in the case of in-plane radiation, the current follows the well-known Friis equation, depending on the characteristics of the junction, dipole antenna, and illumination frequency.

Due to the novelty and clarity of the results and its explanation, this reviewer suggests publication of the work under the premise that the authors address in the manuscript the following comments and questions.

Comments:

1. Figure 1a) shows an artistic representation of the experimental scheme. In there, the representation of the emitted field by the dipole antenna is just in one direction, but according to the simulation showed in Fig. 3a), the radiation must be laterally isotropic.
2. Figure 1c) shows an SEM image with Au regions colored in yellow. Please state that the image is in false color.
3. Figure 2a) and 2b) would better show the information if they are plotted on a logarithmic scale, due to the dependence on $1/d^2$, with d the distance.
3. The ratio E/E_0 shown in Fig. 3d needs to be quantified and mentioned in Line 146 as "minimal".
4. In the Methods/Sample Preparation section, Line 302, it is needed to state the thickness of the Au sacrificial layer in order to make clear the fabrication process.
5. It is necessary to define correctly the optical antenna's diameter because in the Abstract (Line 18), the authors stated 200 nm, while in Line 51, it is 230 ± 10 nm, and in the numerical simulations, it's 220 ± 10 nm (Line 124 and in Methods section Line 299).

Questions:

1. When scanning the sample about the focal region, the step size is 70 nm, i.e. about 3 steps per dipole antenna diameter. Is this resolution sufficient to map correctly the current I_{phot} showed in Fig. 2a) and 2b)?
2. In the reported conditions, what is the minimum current (I_{phot}) that the experimental setup can measure?
3. According to Fig. 1d), the maximum intensity directly illuminating the rectenna was 141 kW/cm^2 , but for remote excitation by a dipole antenna, the intensities were ~ 2.5 times higher, (Fig. 2a, b) and ~ 3.8 , (Fig. 2c). Is it possible to make a comparison with the same intensity for all the cases?

4. What is the efficiency of the dipole antennas, i.e. what fraction of the illumination power is converted into radiative power?

Reviewer #3 (Remarks to the Author):

Summary of the key results:

The manuscript is based on an existing nanostructure geometry which is made of gold, has a sandglass-like shape and features a nanoscale tunnel gap in the middle. By applying a small bias voltage and shining light onto it a photocurrent/-voltage is generated due to the nonlinearity of the tunnel gap [1-2].

The author extend this concept by putting additional gold discs (diameter 230 nm) that act as antennas close to the gap. Individual discs are illuminated, light is scatter from there to the gap region and, hence, a photocurrent current is created again. They verified this effect by using a modulation-demodulation scheme and besides the polarization the authors also vary the disc-gap distance, laser power and offset voltage. All results fit nicely to numerical simulations and analytical modelling.

--

Originality, significance and general opinion:

This is in principle very solid scientific work -- the theory is nicely explained, the experiments are sound and a lot of potential questions are already answered in the supplement information (SI). Furthermore it is easy to read.

However, I am missing a crucial point: The authors apply antennas to an existing design but neglecting that the whole point of using antennas is to enhance emission or absorption of electromagnetic radiation, i.e. they do not compare how the overall device performance improves due the antennas instead of just directly illuminate the gap region. By looking closer at the SI and previous work it seems to me that the performance does not increase due to the antennas but rather decrease and this by nearly one order of magnitude (Fig. S3 vs S4 -- more information would be needed to say that for sure). But this would really question the purpose of the antennas and, hence, the main point of the paper.

Furthermore, the general device performance should be discussed much more extensively within the main text and compared to competing approaches. So far there is only a very brief discussion in the supplement without an actual comparison.

In summary, this is a solid manuscript and from my point of view already worth to publish in a journal. However, for me it would be necessary to clearly show that the antennas improve the overall device performance in order to make it interesting for the audience of Nature

Communication. As long as this cannot be achieved I would suggest publishing it with some minor tweaks (see below) elsewhere.

--

Detailed critiques:

- Fig. 1c should also feature a zoom-in image of the gap in order to get more convinced that it is in the region below 1 nm. (Modern SEMs are capable of delivering such resolutions.)
- The used/cited Simmons paper has an error (the image potential is off by a factor 2) -- see footnote 8 in [3] -- and a corrected model should be used.
- The TiO₂ discussion in S2 is not really convincing and should be extended.
- Line 81 is a bit confusing for the reader as it states $V_{opt} = V_{ac}$ and $I_{phot} = I''$ which is later on not true anymore (Line 89: V_{ac} is adjusted to obtain $I_{phot} = I''$)
- Fit values for Fig. 4b would be nice.

--

References:

- [1] D. R. Ward, F. Hüser, F. Pauly, J. C. Cuevas, and D. Natelson, "Optical rectification and field enhancement in a plasmonic nanogap," *Nat. Nanotechnol.*, vol. 5, no. 10, p. 732, Oct. 2010.
- [2] A. Stolz et al., "Nonlinear Photon-Assisted Tunneling Transport in Optical Gap Antennas," *Nano Lett.*, vol. 14, no. 5, pp. 2330–2338, May 2014.
- [3] J. G. Simmons, "Potential Barriers and Emission-Limited Current Flow Between Closely Spaced Parallel Metal Electrodes," *J. Appl. Phys.*, vol. 35, no. 8, pp. 2472–2481, Aug. 1964.

Reviewer 1:

Reviewer 1: The authors report the use of the optical rectification in an electromigrated nanogap to detect radiation scattered by nearby plasmonic disks. They establish that optical rectification is the most plausible mechanism responsible for the photocurrent through its dependence on dc bias and comparison with (effectively) dc transport, as well as its dependence on the laser position. The authors find that the spatial and polarization dependence of the (dc) photocurrent are consistent with computational modeling of the optical scattering of the plasmonic disks. I find the experimental data and numerical modeling to be quite persuasive. I do have a few comments that I think the authors need to address prior to publication:

1) How many device structures have shown this clear response? That is, unless I'm missing something, it seems that all of the data in the paper and the supporting material come from one particular electromigrated junction. Is that correct? Reproducibility is important, as is establishing the amount of device-to-device variability. One would expect the efficiency of the transduction between excited disk and nanogap to depend sensitively on the nanogap electrical nonlinearity; it would be useful to see this comparison.

Authors: The reviewer is correct. The manuscript is articulated around the characteristics of a single device. The different properties and dependencies of the parameters interrogated necessitate constructing the conclusions based upon the response of a single system. This is to avoid the inconstancy of gathering data from a set of electromigrated junctions. Nonetheless, the point raised by the reviewer is important. Consequently, in the supporting information (**Supplementary Note 4 and Supplementary figure 7**) we present photocurrent map obtain from another device along with its I-V characteristics curve (**Supplementary Figure 3a**). The map is obtained for an illumination intensity 707 kWcm^{-2} and $V_{\text{DC}}=100 \text{ mV}$. This junction also produces a very clear response similar to the rectenna reported in main manuscript albeit with a lower transduction efficiency. As correctly pointed out by the reviewer, it is clear that the rectified photoresponse I_{phot} of a given link is primarily depending on the electrical properties of the electromigrated junction defining the rectenna feedgap. Due to their generally different I-V characteristics, the conversion efficiency of the scattered radiation emitted from the antenna to a transduced current will bear a dependency to the junction's properties. Thus, estimating the fluctuations inherent to the fabrication of the electromigrated junctions provides a quantitative figure to assess device variability. To that aim, we present a statistics of the zero bias conductance of 24 electromigrated junctions in **Supplementary Note 4 and Supplementary Table 1**. Approximately 17% of the electromigrated junctions features a large zero-bias conductance similar to the one reported in the main manuscript. A discussion on reproducibility and device-to-device variability is now included in the **Supplementary Note 4**.

Concerning the point on the non-linearity, we concur with the referee. In case of these electromigrated junctions, nonlinearity in the conductance increases with the applied bias as can be seen in the I-V characteristics displayed for instance in **Fig. 1c** and **Supplementary Fig. 3**. Therefore higher applied bias would mean an increased transduction yield, which is the message carried by **Fig. 5a** of the revised version of the manuscript. We now include the experimental quantification of this effect in the “**Controlling the efficiency**” section Under **Results and Discussion** in the main manuscript. We also mention in the main manuscript that raising the applied bias across the junction V_{dc} increases the transduction yield from -91dBm to -87dBm for the signal transmitted from the antenna located $2 \mu\text{m}$ away from the rectenna.

Reviewer 1: 2) The authors have already performed another cross-check that measured photocurrent response is not dominated by some thermal response. The absorption and hence heating of the illuminated disks should be insensitive to the incident polarization. The fact that the photocurrent does have polarization dependence (consistent with the modeling), while heating due to the disks should be largely polarization independent, is additional evidence in favor of the authors' interpretation of the mechanism.

Authors: Yes, we agree with the reviewer. The effect of the polarization together with the other analysis provided in the supplementary information is strong evidence supporting the rectification picture.

Reviewer 1: 3) The authors should say something in the methods or the supporting material about how they performed the finite element simulations. What software was used? What dielectric functions were assumed for the metal? Did they include the Ti or TiO_x in the calculations? Was the incident excitation modeled as a gaussian beam, or was the incident excitation modeled as a plane-wave? I would think the former, but it is important to make this clear to the reader.

Authors: In the revised manuscript we include the description of the numerical modeling in the **Methods** section under **Numerical modeling segment**. We used a commercial FEM package from Comsol Multiphysics to perform the simulations. The dielectric function of gold is taken from experimental values provided by Johnson and Christy. In the experiment, a 2 nm thick Ti layer is used for promoting adhesion between gold and the SiO₂ substrate. In the numerical simulations, we did not include the Ti layer. While such layer is usually considered detrimental from the performance point of view of a plasmonic material, the purpose of the simulation was here to understand the physical mechanisms, in particular the effect of polarization. The figures provided by the simulations may indeed be affected by the presence of the Ti layer, but the overall mechanism remains qualitatively the same. Nonetheless, we have refined the simulation in the revised manuscript to model a system closer to the experimental reality. The original manuscript included calculations considering the incident excitation as a localized plane wave source on the gold nanodisk with a spot size of 1 μm diameter. In the revised version, by taking the reviewer's suggestion, we performed all the numerical calculations again by considering a focused Gaussian beam of 600 nm diameter spot size, corresponding to a full-width at half-maximum of ~350 nm.

Reviewer 1: 4) It would be helpful for the authors to calculate an efficiency of the coupling process. The optical rectification in these nanogap structures has historically had a very poor efficiency in terms of power. If there really is to be an application of similar structures in on-chip signal processing, efficiency is important to quantify.

Authors: This is a good point. We calculate the power transfer efficiency between the optically excited nano antenna and the receiving rectenna in the form of a transduction yield, which is a measure of how much of the fed optical power is transduced as electrical power in the circuit (external efficiency). We find out that with our device a transduction yield of approaching -80 dBm is achievable. This is a low figure, but it is interesting to benchmark that number against the widely used RF wireless interconnects. In RF wireless communication the quality of the transmitted signal is evaluated by the received signal strength indicator (RSSI) measured at the reception node *before* any transduction of the RF wave to an electrical signal. A transmission channel with RSSI comprised between -70 dBm to -100 dBm is considered as fair. Here, we obtain an electrical power in the range of -85 to -101 dBm w.r.t the incident optical power *after* optical-to-electrical transduction. A detailed discussion has been included in the main text in the revised manuscript in the paragraph with the heading **Transduction yield** under the subsection **controlling the efficiency of the wireless link** in section **Results and Discussion**.

Reviewer 1: 5) I would suggest that the authors qualify some of the statements made in the last few sentences of the paper. For example, better to say something like "in principle, data rates as high as 10¹² bps could be possible, as the ultimate limiting factor is the polarization response of the metal". (Note also that the authors must mean 10¹² bps, not 10¹² Mbps!) Also, the statement that the enhanced field in the

nanogap is appropriate for driving transistor operations seems not to be well motivated. How would that work? Better to leave that out unless there is some clear reasoning.

Authors: We thank the reviewer for pointing out the typo of 10^{12}Mbs^{-1} . In the revised version of the manuscript it has been corrected to 10^{12}bs^{-1} along the lines suggested by the reviewer.

Reviewer 1: *In summary, I find the data and interpretation to be very interesting and persuasive. I think that the paper could be quite influential in the field, but it is important to establish reproducibility and to provide further quantification of efficiency, as well as greater information about the numerical modeling.*

Authors: We thank the reviewer for this positive appraisal of our work. We sincerely hope to have responded satisfactorily to her/his queries about reproducibility, efficiency and required details.

Reviewer 2

Reviewer 2: *The authors report on a configuration of energy transmission via an in-plane wireless link and the detection by electrical rectification in an atomic-scale tunneling junction. A systematic analysis of the device is presented on the current generated when the junction is illuminated directly by out-of-plane and in-plane sources. Interestingly, in the case of in-plane radiation, the current follows the well-known Friis equation, depending on the characteristics of the junction, dipole antenna, and illumination frequency. Due to the novelty and clarity of the results and its explanation, this reviewer suggests publication of the work under the premise that the authors address in the manuscript the following comments and questions.*

General comments

Reviewer 2: 1. *Figure 1a) shows an artistic representation of the experimental scheme. In there, the representation of the emitted field by the dipole antenna is just in one direction, but according to the simulation showed in Fig. 3a), the radiation must be laterally isotropic.*

Authors: We thank the reviewer for pointing this out. **Figure 1a** has been modified accordingly in the revised manuscript.

Reviewer 2: 2. *Figure 1c) shows an SEM image with Au regions colored in yellow. Please state that the image is in false color.*

Authors: The reviewer is correct; the image is colored to discriminate the material constituting the device. In the revised manuscript, we mention that the SEM image is in false colours and the yellow hue designates Au.

Reviewer 2: 3. *Figure 2a) and 2b) would better show the information if they are plotted on a logarithmic scale, due to the dependence on $1/d^2$, with d the distance.*

Authors: We believe the reviewer refers to Fig. 4a and 4b of the original manuscript. Although we agree with the reviewer that the information would be clearer if plotted on a logarithmic scale, the ever-present noise current in the measurement system do not allow us do so. In **Results and Discussion** under the section **Controlling the efficiency of the wireless link** we mention the presence of a noise current of 0.32 nA. Therefore, we fit the experimental data with a generic power law of ad^b+c where c signifies the noise level. Due to this term c in the fitting parameter, simply plotting I_{phot} vs d in logarithmic scale would not result in a linear fit with slope of -2.

Reviewer 2: 3. The ratio E/E_0 shown in Fig. 3d needs to be quantified and mentioned in Line 146 as “minimal”.

Authors: The ratio E/E_0 at the junction in case of figure 3d-f where the optical antenna is illuminated with incident polarization along x-axis is 0.44. This is now mentioned in the manuscript. For clarity in the electric field map presented in Figure 3f, the calculated field is multiplied by a factor 5 and plotted with the colour scale range between 0 and 5.6.

Reviewer 2: 4. In the Methods/Sample Preparation section, Line 302, it is needed to state the thickness of the Au sacrificial layer in order to make clear the fabrication process.

Authors: Taking the suggestion into account, we mention that the thickness of the sacrificial gold layer is 10-15nm in the **Sample preparation** in the **Methods** section. This layer is removed during the development step done after the electron exposure.

Reviewer 2: 5. It is necessary to define correctly the optical antenna's diameter because in the Abstract (Line 18), the authors stated 200 nm, while in Line 51, it is 230 ± 10 nm, and in the numerical simulations, it's 220 ± 10 nm (Line 124 and in Methods section Line 299).

Authors: We thank the reviewer for pointing out these typos. The nanodisks have diameter of 220 ± 10 nm and the numerical modeling is performed assuming a 220 nm diameter disk. This has been corrected in the revised manuscript.

Questions:

Reviewer 2: 1. When scanning the sample about the focal region, the step size is 70 nm, i.e. about 3 steps per dipole antenna diameter. Is this resolution sufficient to map correctly the current I_{phot} showed in Fig. 2a) and 2b)?

Authors: The lateral resolution of the instrument is dictated by the point-spread function of the instrument. To a fairly good approximation, the Rayleigh criterion $0.6\lambda/NA$ provides an estimate of this parameter. Where $\lambda=785$ nm is the illumination wavelength and NA is the numerical aperture of the objective (1.49). The point-spread function of the microscope is thus approximately 320 nm. Shannon-Nyquist theorem tells us that the minimum resolution should be discretized by 2 pixels. However, in general confocal practice, scans tend to be slightly oversampled to render better quality images, even if there is no additional gain of spatial information. In the present case, we use an oversampling ratio of 4.5.

Reviewer 2: 2. In the reported conditions, what is the minimum current (I_{phot}) that the experimental setup can measure?

Authors: The noise level for measuring I_{phot} lies around 320 pA. Therefore, it is the minimum value of I_{phot} that can be measured experimentally in the current scenario. We specified that value in the revised manuscript.

Reviewer 2: 3. According to Fig. 1d), the maximum intensity directly illuminating the rectenna was 141 kW/cm², but for remote excitation by a dipole antenna, the intensities were ~2.5 times higher, (Fig. 2a, b) and ~3.8, (Fig. 2c). Is it possible to make a comparison with the same intensity for all the cases?

Authors: In the supplementary figure S4c, where we characterize the thermal contributions in the photocurrent response, we present a I_{phot} map for an incident laser intensity of 353 kW/cm² and $V_{\text{dc}}=50$ mV. Under this operating condition, we record a rectified photocurrent of 10.3 nA for a direct excitation of the nanojunction. This is approximately 6 times higher than the observed rectified current (1.67 nA) when the 2 μm distant antenna is excited. In **Supplementary Table 2**, we include a comparison of the currents measured by rectifying the signal transmitted by the different antennas with a direct rectification from the junction.

Reviewer 2: 4. What is the efficiency of the dipole antennas, i.e. what fraction of the illumination power is converted into radiative power?

Authors 2: This is a good question. To address this point, we use a computational approach to quantify the amount of power effectively radiated by a given antenna. We numerically calculate the efficiency of conversion of radiative power from the illumination power by calculating total scattered power from the illuminated nano-antenna and by normalizing it to the incident power. We present a graph representing the variation in the scattering efficiency with the diameter of the antenna in the **Supplementary Figure 8a** and **b** in the revised version of the supplementary information.

Reviewer #3

Reviewer 3: The manuscript is based on an existing nanostructure geometry which is made of gold, has a sandglass-like shape and features a nanoscale tunnel gap in the middle. By applying a small bias voltage and shining light onto it a photocurrent/-voltage is generated due to the nonlinearity of the tunnel gap [1-2]. The author extend this concept by putting additional gold discs (diameter 230 nm) that act as antennas close to the gap. Individual discs are illuminated, light is scatter from there to the gap region and, hence, a photocurrent current is created again. They verified this effect by using a modulation-demodulation scheme and besides the polarization the authors also vary the disc-gap distance, laser power and offset voltage. All results fit nicely to numerical simulations and analytical modelling.

I am missing a crucial point: The authors apply antennas to an existing design but neglecting that the whole point of using antennas is to enhance emission or absorption of electromagnetic radiation, i.e. they do not compare how the overall device performance improves due the antennas instead of just directly illuminate the gap region. By looking closer at the SI and previous work it seems to me that the performance does not increase due to the antennas but rather decrease and this by nearly one order of magnitude (Fig. S3 vs S4 -- more information would be needed to say that for sure). But this would really question the purpose of the antennas and, hence, the main point of the paper. Furthermore, the general device performance should be

discussed much more extensively within the main text and compared to competing approaches. So far there is only a very brief discussion in the supplement without an actual comparison.

Authors: We do not expect an improved performance by transmitting the signal via the mediation of antenna compared to a direct excitation of the rectenna. The transduction will always be better when the rectenna is directly illuminated compared to a remote excitation promoted by scattering. The important point however, is that our approach enables a communication between two distant functional nano-devices, and this is the purpose of our work. Our aim for developing such wireless platform is to trigger a new paradigm for on-chip data transfer. Current technologies are essentially based on physical links (i.e. waveguides) to transport optical information. Nano-scale integration is limited by the size of the waveguiding structures, as well as by cross-talks issues. Our work may open a new approach for micrometer-scale routing in future highly integrated opto-electronic circuits by mitigating the deployment of photonic waveguides where short interconnects will be required. To provide a complete response, we argue along the following argumentation. Metal antennas are widely deployed to enhance the interaction of light, especially when excited at the surface plasmon resonance. For instance, such capability is at the heart of molecular plasmonics, where a desired response is greatly amplified by a *near-field* interaction between the molecule and the plasmonic nanostructure. In the configuration here, the generated photocurrent results from the propagation of the homogenous field components radiated by the dipolar response of the antenna, i.e. near-field interactions at the transmitting element are irrelevant. The important parameter, which dictates the performance of the link, is the scattering efficiency defining how much incident radiation received by the antenna can be transmitted to far-field and hence to the rectenna. For the optical link discussed here, where the signal is transferred from a transmitter to a receiver, the purpose of the antenna is to direct the incoming radiation towards the rectenna as efficiently as possible. This can be leveraged by different strategies: a simple approach is to rely on the specifics of the radiation pattern when the dipolar plasmonic mode of the antenna is excited and/or to adjust the antenna diameter to increase its scattering cross-section. We used this route to transmit the signal. There are other alternatives to improve the performance. For instance, one may design highly directive antennas (e.g. Yagi-Uda), which would concentrate the radiation transmitted to a restricted angular region directed towards the rectenna. In the supplementary material, we included a detail discussion (**Supplementary note 5** and **Supplementary Figure 8**) on the dependence of the scattering efficiency with the diameter of the transmitting antenna. We also provide in **Supplementary Table 2**, a comparison of the rectified current measured from a direct illumination of the junction with a signal scattered by the distant antennas.

Reviewer 3: *Furthermore, the general device performance should be discussed much more extensively within the main text and compared to competing approaches. So far there is only a very brief discussion in the supplement without an actual comparison.*

Authors: We agree with the reviewer. We complemented the discussion about the critical parameters affecting the performance of the link. In particular, we moved in the revised manuscript the effects of a smaller antenna and larger gap size. We also add a discussion about the efficiency of the link, which includes the effect of scattering, signal transmission and transduction. We compare the values with standard RF wireless link. While, the latter cannot be considered as a competing approach, we think it is interesting to benchmark the figures. We include in the supplementary (**supplementary Figure 8**) an extended discussion about the scattering efficiency of the antenna and the comparison between a direct illumination of the rectenna with a transmission mediated via the wireless link (**Supplementary Note 5** and **Supplementary Table 2**).

Reviewer 3: *In summary, this is a solid manuscript and from my point of view already worth to publish in a journal. However, for me it would be necessary to clearly show that the antennas improve the overall device performance in order to make it interesting for the audience of Nature Communication. As long as this cannot be achieved I would suggest publishing it with some minor tweaks (see below) elsewhere.*

Authors: We thank the reviewer for this positive comment. To reiterate the response given above, the main purpose of the antenna is to redirect the incoming laser signal towards the rectenna. The concept is to provide a technological step enabling on-chip nanoscale wireless broadcasting of information at optical frequency. Short-scale optical communication between remote nanodevices is a necessity for cases where space constraint is preventing implementation of physical links or where three-dimensional optical information transfer is required. The antenna performance is thus cast in its radiation pattern rather than in its near-field properties.

Detailed critics:

Reviewer 3 *Fig. 1c should also feature a zoom-in image of the gap in order to get more convinced that it is in the region below 1 nm. (Modern SEMs are capable of delivering such resolutions.)*

Authors: We understand the position of the reviewer. 0.6 to 0.8 nm high-resolution imaging of very specific samples may be achievable with a particularly optimized SEM microscope operating in a quiet electrical and vibrational environment. According to our simulation of the transport, the gap size is about 5 Å, which is at the limit of best microscopes. This is certainly not the case with our SEM located in a noisy clean room. Imaging a pristine gap with a high resolution SEM microscope proves to be extremely challenging because our devices are fabricated on a non-conductive glass substrate. This precludes operating the SEM under optimum acceleration voltages because charging effects disturbs the image. In the SEM micrograph presented in the manuscript, the image was obtained by sputtering the device with a thin layer of Au to enable the evacuation of the charges. The procedure is thus destructive. Finally, we believe that a high-resolution image may indeed be a valuable addition to confirm the size of the gap deduced from Simmons' model of the transport characteristics. However, the morphological complexity of an electromigrated gap is such that a definitive assessment would anyway be disputable. We have added a discussion in the supplemental information to take into account that point (**Supplementary Note 1**).

Reviewer 3: *The used/cited Simmons paper has an error (the image potential is off by a factor 2) -- see footnote 8 in [3] -- and a corrected model should be used.*

Authors: Yes, the referee is correct. There is an error in deriving the image potential in the Simmons paper. However, the missing factor does not affect our modeling. In our model of the electrical characteristics we use the generalized formula for calculating J-V curve provided in these papers and estimate the gap width, average barrier height and the difference between the barrier heights on the two sides of the gap. In all the Simmons papers, a model of the image potential is used to estimate the change in the *average* barrier height (ϕ) and then that average value is inserted in the generalized formula (which does not change its form) to reconstruct the J-V curve. By our model we directly derive the average barrier height, which includes all the effects including shortening of the barrier due to the image potential.

Reviewer 3: The TiO2 discussion in S2 is not really convincing and should be extended.

Authors: Following the suggestion, we extend the discussion with help of a Fowler-Nordheim plot (see **Supplementary Figure 3**) of the I-V characteristics of an electromigrated gold junction in the revised version of the supplementary information. In particular, we address this question by analyzing the potential role of the oxide on the electrical characteristics, notably the energy barrier experienced by the tunneling electrons and the interpretation of the Fowler-Nordheim representation. The minimum of the F-N plot is subject to an active debate in the community. The latest literature suggests that the transition voltage is not only depending on the barrier height, but is also inversely dependent on the size of the gap. We write an extended discussion together with corresponding illustrations in **supplementary Note 1**. To make our point stronger, we provide for reviewing purpose a graph showing the linear dependence of experimentally determined transition voltages measured on the positive and negative sides of the F-N plot with $(\phi)^{-1/2}g^{-1}$, where ϕ is the barrier height and g is the gap width. Both parameters are deduced from fitting the I/V characteristics of a series of electromigrated junctions using Simmons' equation. We have not included this graph in the supplementary material, as it will be subject to a subsequent manuscript.

Figure 1: Evolution of the transition voltages evaluated from the Fowler-Nordheim plots with $(\phi)^{-1/2}g^{-1}$. The solid line is the expected value of V_T .

Reviewer 3: Line 81 is a bit confusing for the reader as it states $V_{opt} = V_{ac}$ and $I_{phot} = I''$ which is later on not true anymore (Line 89: V_{ac} is adjusted to obtain $I_{phot} = I''$)

Authors: Yes, we agree with reviewer. We rephrase the line in the revised manuscript to avoid confusions.

Reviewer 3: Fit values for Fig. 4b would be nice.

Authors: We thank the reviewer for the suggestion. The slopes of the linear fits actually signify the coupling efficiency for each antennas and the rectenna and we present it in form of table in the revised manuscript. Also, note that the **figure 4b** is now **figure 5b** in revised version.

REVIEWERS' COMMENTS:

Reviewer #1 (Remarks to the Author):

The edits and additions made by the authors in response to the referee reports have improved the clarity and readability of the manuscript, and added important information for other investigators. I support publication. A couple of very minor points:

Page 8, line 175 - better to say "When the dipolar radiation is polarized perpendicularly to the receiver".

At the end of supplementary note 1, better to say "...is not sufficient to determine the presence [or role] of a TiO₂ layer" (new words in brackets).

Beginning of supplementary note 2, better grammar to say "...receives radiation" rather than "receives a radiation".

I still think it's worth adding a sentence that explicitly points out: Since the absorption of the disk antenna is independent of incident polarization, any optical heating-based transduction mechanism would also be expected to be polarization-independent. The experimentally observed polarization dependence removes any concern that the transduction mechanism might be based on some thermal deformation process rather than optical rectification.

Reviewer #2 (Remarks to the Author):

The authors have satisfactorily responded to the criticism and questions of this reviewer, and have included additional detail in the main ms and in the supplemental material. These additions have significantly improved the clarity of the results and their interpretation, and have elevated the quality of the paper to the point of meriting a recommendation of acceptance.

Reviewer #3 (Remarks to the Author):

The authors have carefully addressed all points the reviewers raised and subsequently also updated the manuscript.

My main concern in the last round was that they do not show that the antennas improve the overall device performance. With their reply they were able to convince me that although for this proof-of-principle experiment with direct laser excitation from the top this might be true, for applications with in-plane light propagation the antennas will play a key role for enhancing the collection efficiency. I am satisfied with this explanation but think that it should also be clearly pointed out in the main text, e.g. in the introduction, in order to let the reader better understand the limitations.

Furthermore, as for the illumination-from-the-top situation the antennas show now advantage I would be more careful when advertising inter chip interconnects and 3D integrated circuits as future applications, as readers might think of glass fiber from the top or similar scenarios in these cases.

Besides these issues all my other concerns were answered appropriately and I think the questions from the other reviews, too, such that I would recommend the paper to be published with the above minor changes.

As a final note I would recommend to not use the letter "phi" for the barrier height AND the transduction yield.

Reviewer 1:

Reviewer 1: The edits and additions made by the authors in response to the referee reports have improved the clarity and readability of the manuscript, and added important information for other investigators. I support publication.

A couple of very minor points:

Page 8, line 175 - better to say "When the dipolar radiation is polarized perpendicularly to the receiver".

At the end of supplementary note 1, better to say "...is not sufficient to determine the presence [or role] of a TiO₂ layer" (new words in brackets).

Beginning of supplementary note 2, better grammar to say "...receives radiation" rather than "receives a radiation".

Authors: We thank the reviewer for these suggestions. We revised the manuscript accordingly.

Reviewer 1: I still think it's worth adding a sentence that explicitly points out: Since the absorption of the disk antenna is independent of incident polarization, any optical heating-based transduction mechanism would also be expected to be polarization-independent. The experimentally observed polarization dependence removes any concern that the transduction mechanism might be based on some thermal deformation process rather than optical rectification.

Authors: We agree on that point. We modified a paragraph at the end of section "Optical wireless link" to take into account this remark. The sentences read "*This polarization dependence of the current produced by the rectenna removes any concern about a heat-mediated transduction mechanism since the temperature of the illuminated disc antenna is driven by the absorption coefficient and not by the polarization of the incident laser beam. As an additional control experiment we map dI/dV at the frequency of the optical chopper (Supplementary Figure 5). No measurable contrast can be related to the photocurrent maps presented in Fig. 2a (more discussion in Supplementary Note 2) indicating that the gap size is not deformed by the heat produced at the distant antenna*"

Reviewer 2:

The authors have satisfactorily responded to the criticism and questions of this reviewer, and have included additional detail in the main ms and in the supplemental material. These additions have significantly improved the clarity of the results and their interpretation, and have elevated the quality of the paper to the point of meriting a recommendation of acceptance.

Authors: Thanks to the suggestions and critics raised by the reviewing process, we are delighted to see our revised manuscript positively evaluated.

Reviewer 3:

Reviewer 3: The authors have carefully addressed all points the reviewers raised and subsequently also updated the manuscript.

My main concern in the last round was that they do not show that the antennas improve the overall device performance. With their reply they were able to convince me that although for this proof-of-principle experiment with direct laser excitation from the top this might be true, for applications with in-plane light propagation the antennas will play a key role for enhancing the collection efficiency. I am satisfied with this explanation but think that it should also be clearly pointed out in the main text, e.g. in the introduction, in order to let the reader better understand the limitations.

Authors: We concur with the reviewer. We have reformulated the introduction and discuss the limitation in the final section of the manuscript. In particular we added in the discussion that a wireless link might be valuable not only when the dimension of waveguides are constraining the integration, but also when the information should be relayed out of a guided architecture by the use of optical antenna and arrays. We added references [29-30] in this context.

Reviewer 3: Furthermore, as for the illumination-from-the-top situation the antennas show now advantage I would be more careful when advertising inter chip interconnects and 3D integrated circuits as future applications, as readers might think of glass fiber from the top or similar scenarios in these cases.

Authors: This statement is not entirely correct. The manuscript is essentially based on the beneficial far-field radiation properties of the antenna! On one hand, the polarization-sensitive dipolar emission pattern is clearly helping us to exclude thermal effects. This would have been considerably more difficult with a unit scattering the field in no preferred direction. In this respect, the antenna is of clear advantage. On the other hand, we show in the Supplementary Figure 8 that the amount of scattered power by the antenna may be improved when working with resonant antennas. Even though the scattering power increases with diameter, we think that for integration-driven on-chip applications constraining the antenna footprint to nanoscale dimensions resonant scattering devices will thus be a necessity. We think this is a second advantage, even though in the proof-of-principle experiment discussed here, we certainly did not fully used the benefit of an optimized plasmonic antennas with (i) highly directional emission pattern and (ii) enhanced interaction cross-section. Some elements of this discussion were already included in the section “discussion”. To go along the suggestion of the reviewer, we removed a sentence in the concluding section that may be indeed controversial at the current stage of the technological development: “Therefore, it can be harnessed as inter and intra chip interconnects for network-on-chips, 3 dimensional integrated circuits and high speed connectors.”

Reviewer 3: As a final note I would recommend to not use the letter “phi” for the barrier height AND the transduction yield.

Authors: Yes, indeed! We now introduce the Greek letter κ for the transduction yield in the revised manuscript.